# Fatigue following type 2 diabetes: Psychometric testing of the Indonesian version of the multidimensional fatigue Inventory-20 and unmet fatigue-related needs

**Debby Syahru Romadlon**[1], **Hui-Chuan Huang**[1], **Yang-Ching Chen**[2,3], **Sophia H. Hu**[4], **Faizul Hasan**[1], **Milton D. Chiang Morales**[5], **Ollyvia Freeska Dwi Marta**[6], **Safiruddin Al Baqi**[7], **Hsiao-Yean Chiu**[1,8,9] *

1 School of Nursing, College of Nursing, Taipei Medical University, Taipei, Taiwan, 2 Department of Family Medicine, School of Medicine, College of Medicine, Taipei Medical University, Taipei, Taiwan, 3 Department of Family Medicine, Taipei Medical University Hospital, Taipei, Taiwan, 4 Department of Nursing, College of Nursing, National Yang Ming Chiao Tung University, Taipei, Taiwan, 5 Graduate Institute of Clinical Medicine, College of Medicine, Taipei Medical University, Taipei, Taiwan, 6 Department of Nursing, Faculty of Health Science, University of Muhammadiyah Malang, Malang, Indonesia, 7 Institut Agama Islam Negeri Ponorogo, Ponorogo, Indonesia, 8 Research Center of Sleep Medicine, College of Medicine, Taipei Medical University, Taipei, Taiwan, 9 Department of Nursing, Taipei Medical University Hospital, Taipei, Taiwan

* hychiu0315@tmu.edu.tw

## Abstract

Patients with type 2 diabetes mellitus (T2DM) often experience fatigue. The Multidimensional Fatigue Inventory (MFI-20) is a valid tool for evaluating fatigue; however, its psychometric properties have not been examined in Indonesian-speaking patients with T2DM. This study assessed the psychometric properties of the Indonesian version of the Multidimensional Fatigue Inventory-20 (IMFI-20) in patients with T2DM and investigated fatigue in a health-care setting. A cross-sectional design was adopted. Two hundred patients with T2DM were interviewed in clinics. Five self-structured measures were used to assess the frequency and duration of fatigue and the health-care utilization of patients with fatigue. Cronbach's alpha and intraclass correlation (ICC) were used to evaluate the internal consistency and test–retest reliability of the Indonesian version of the MFI-20 (IMFI-20). The criterion, convergent, and known-group validity of the IMFI-20 were also examined, and its underlying structure was determined using explanatory factor analysis. The STROBE checklist was used. The results revealed that approximately half of the patients experienced fatigue. Among those with fatigue, 62% reported that their fatigue was rarely or never treated by their physicians. The IMFI-20 exhibited satisfactory model fit, excellent internal consistency (Cronbach's alpha of 0.92), and test–retest ICC of 0.93. The IMFI-20 was significantly associated with the Functional Assessment of Chronic Illness Therapy–Fatigue, Beck Depression Inventory-Second Edition, and Pittsburgh Sleep Quality Index ($r = 0.705$, 0.670, and 0.581, respectively). The IMFI-20 exhibited known-group validity for unfavorable sleep quality and HbA1C $\geq$ 6.5%. Our findings suggest that patients with T2DM who experience fatigue are often overlooked by health-care providers, and that the IMFI-20, which

**Data Availability Statement:** All relevant data are within the paper and its Supporting information files.

**Funding:** Ministry of Science and Technology, Taiwan (MOST 111-2628-B- 038-008 and MOST 111-2314-B-038-033-MY3).

**Competing interests:** The authors have declared that no competing interests exist.

exhibits excellent psychometric properties, can be adopted by studies that use fatigue as an endpoint in Indonesian-speaking populations.

## 1. Introduction

Fatigue, which can be defined as the subjective perception of overwhelming and debilitating weakness, lack of energy, or tiredness [1], is a common and debilitating symptom experienced by numerous patients with type 2 diabetes mellitus (T2DM); its prevalence in this population is estimated to be 50% [2]. Fatigue can be triggered by an individual's lifestyle and various medical, psychological, and glycemia/diabetes-related factors [3], and it is associated with overweight [4], inflammation [5], glycemic variability [6], reduced diabetes self-care ability [7], sleep problems [8], reduced quality of life, and impaired functional status [4]. Despite its high prevalence and substantial effects, fatigue in individuals with T2DM is often neglected by health-care providers. Therefore, the use of a valid and reproducible instrument to evaluate fatigue in individuals with T2DM is essential for future clinical practice and research.

Fatigue is a multidimensional construct with physiological, emotional, and mental components [9]. To date, several self-administered measures can be used to evaluate fatigue in individuals with diabetes; they include the Functional Assessment Chronic Illness Therapy (FACIT)–Fatigue scale [10], Fatigue Severity Scale, Fatigue Assessment Scale, and Visual Analog Fatigue Scale [11]; however, these tools are structured unidimensionally, such that they cannot adequately assess the multidimensional aspects of fatigue in patients with diabetes. By contrast, the Multidimensional Fatigue Inventory-20 (MFI-20), which was developed by Smets, enables a thorough assessment of patient perception and a comprehensive understanding of their physical and mental fatigue [12]. The MFI-20 comprises five subscales (i.e., general fatigue, physical fatigue, reduced activity, reduced motivation, and mental fatigue), and it has been demonstrated to exhibit high reliability and validity [12, 13]. Because of its high utilizability, the MFI-20 has been translated into multiple languages and has been applied to various populations, including patients with cancer, Parkinson's disease, coronary artery disease, chronic hepatitis B, thyroid disease, inflammatory bowel disease, and major depression [14–25]. However, no study has investigated the psychometric properties of the MFI-20 in the context of Indonesian-speaking patients with diabetes.

Several cross-cultural translational studies have reported that the MFI-20 should comprise no more than three to four dimensions when applied to populations that speak multiple languages and live with various chronic illnesses [14, 16, 26]. Because the MFI has not been localized for Indonesian-speaking populations or applied to populations with T2DM, we translated the English version of the MFI-20 into Indonesian to establish the Indonesian version of the MFI-20 (IMFI-20) and investigated its reliability and validity for patients with T2DM. In addition, we explored its use for assessing the fatigue of Indonesian-speaking individuals with T2DM.

## 2. Materials and methods

### 2.1 Study design and settings

This study adopted a prospective methodological design with convenience sampling, and it was conducted across three diabetes management centers in Malang City, East Java, Indonesia, from October 2021 to February 2022. The study was approved by the Joint Institutional Review Board of Taipei Medical University (No. N202109026) and the Health Research Ethics

Committee Institute of Health Science STRADA Indonesia (No. 2395/KEPK/VII/2021). Written consent was obtained from participants who agreed to participate in the survey. In addition, this study was reported in accordance with the Strengthening the Reporting of Observational Studies in Epidemiology (i.e., STROBE) guidelines [27] (S1 Table).

## 2.2 Study population

In this study, individuals who were diagnosed with T2DM and aged 17 years or more (the legal age at which informed consent can be provided in Indonesia) were enrolled as participants. Individuals were excluded if they were incapable of reading and writing Indonesian; were diagnosed with cognitive impairment, psychiatric diseases, substance abuse, or sleep disorders; or were diagnosed with cancer and undergoing treatment prior to the study. The estimation of the sample size was based on the ratio of the number of people ($N$) to the number of measured variables ($p$) [28–30]; the widely accepted ratio of 10 cases per indicator variable was adopted. Because the IMFI-20 contains 20 items (variables), a sample size of 200 was required.

## 2.3 Translation process

From Smets (the developer of the MFI-20), we obtained permission to translate the MFI-20 into Indonesian. Translation was performed in accordance with the guidelines of Beaton, Bombardier, Guillemin, and Ferraz [31]. First, three translators who were native speakers of Indonesian and proficient in English independently translated the English version of the inventory into Indonesian. Next, two experts (a nursing lecturer and an English lecturer) discussed and analyzed the differences among the three Indonesian versions and subsequently merged these versions to create a single complete translation. Two native English–speaking back-translators who were proficient in Indonesian then independently translated the completed translation into the source language (English). Finally, two nursing experts and a physician discussed and analyzed the back-translated version, after which the Indonesian version of the IMFI-20 was finalized.

## 2.4 Instruments

**2.4.1 IMFI-20.** The IMFI-20 comprises 20 items and five subscales (i.e., general fatigue, physical fatigue, reduced motivation, reduced activity, and mental fatigue) [12]. Each subscale contains four items that are rated on a 5-point Likert scale ranging from 1 (*strongly agree*) to 5 (*strongly disagree*). Ten positively worded items were reverse-scored (i.e., items 2, 5, 9, 10, 13, 14, 16, 17, 18, and 19). The scores of each subscale (range = 4–20) were calculated as the sum of the corresponding item scores, and the total fatigue score (range = 20–100) was calculated as the sum of the subscale scores; a higher score indicated a higher level of fatigue. Strong validity and acceptable reliability have been demonstrated for the original MFI-20 (Cronbach's alpha = 0.84) [12].

**2.4.2 FACIT–Fatigue scale.** To assess the convergent validity of the IMFI-20, fatigue was also investigated using the FACIT–Fatigue scale [11]. The scale comprises 13 items for measuring fatigue and tiredness and their effects on daily activities and function in relation to various chronic illnesses. Each item is rated on a response scale, ranging from 0 (*not at all*) to 4 (*very much so*). A total score between 0 and 52 is obtained by summing all items, and a higher score indicates less fatigue [11]. The FACIT–Fatigue scale exhibited strong validity and reliability for patients with cancers and other chronic diseases [11, 32], and scholars have used it to investigate the fatigue level in patients with T2DM [10]. Acceptable reliability has been demonstrated for the Indonesian version of the FACIT–Fatigue scale (Cronbach's alpha = 0.646) [33]; for our study population, it exhibited excellent internal consistency (Cronbach's alpha = 0.91).

**2.4.3 Beck depression inventory–second edition.** The Beck Depression Inventory–Second Edition (BDI-II) comprises 21 items for measuring subjective depression symptoms in the preceding 2 weeks [34]. Each item is scored on a 4-point Likert scale ranging from 0 to 3, and the total score of the BDI-II ranges from 0 to 63 [34]. The BDI-II comprises cognitive, emotional, and somatic components [35]. The Indonesian version of the BDI-II (Indonesian BDI-II) was developed for the general Indonesian-speaking population and for Indonesian-speaking patients with coronary heart disease [36]. Acceptable convergent validity and high internal consistency have been demonstrated for the Indonesian BDI-II (Cronbach's alpha = 0.90) [36], and in our study, its Cronbach's alpha was 0.88 in patients with T2DM.

**2.4.4 Pittsburgh Sleep Quality Index.** The Pittsburgh Sleep Quality Index (PSQI) assesses self-reported sleep quality and sleep disturbance in the preceding month. The scale comprises 19 items and the seven following dimensions: (1) subjective sleep quality, (2) sleep latency, (3) sleep duration, (4) sleep efficiency, (5) sleep disturbance, (6) sleeping medication use, and (7) daytime dysfunction. Each item is rated on a 4-point Likert scale ranging from 0 to 3, and the overall score is between 0 and 20. A score of <5 indicates favorable sleep quality [37]. The Indonesian version of the PSQI exhibited high validity and reliability (Cronbach's alpha = 0.72) in the adolescent population [38]. The PSQI exhibited acceptable internal consistency (Cronbach's alpha of 0.78) in our study population.

**2.4.5 Demographic and disease characteristics and self-reported fatigue in health-care settings.** A predesigned information sheet was used to collect demographic and disease characteristics, including age, gender, body mass index, education level, marriage status, income level, current treatment for diabetes, blood glucose, and HbA1C level, all of which are associated with fatigue in individuals with T2DM [2]. In addition, we designed five questions; two questions pertained to whether an individual experienced fatigue and the duration of the fatigue experienced (question 1, *Do you feel fatigue*?; question 2, *How long have you felt fatigue*?), and the remaining three questions were scored on a 5-point Likert scale ranging from 0 (*never*) to 4 (*always*) (question 3, *Have you ever discussed "the feeling of tiredness or fatigue" with your physician*?; question 4, *Has your fatigue level been measured by your physician*?; question 5, *Has your fatigue been treated by a physician*?). These questions were used to assess the fatigue of individuals with T2DM in a health-care setting.

## 2.5 Statistical analysis

All analyses were performed using SPPS version 24.0 (IBM, Armonk, NY, USA), and a *P* value of <0.05 indicated statistical significance.

**2.5.1 Descriptive analysis.** Descriptive analyses were performed to evaluate the distributions, floor effects, and ceiling effects of the total and subdomain scores of the IMFI-20. For a comparison of demographic characteristics and fatigue scores, the independent *t* test was used for evaluating continuous variables and the chi-squared test for categorical variables.

**2.5.2 Reliability, floor effect, and ceiling effect.** The Cronbach's alpha values of the IMFI-20, its subscales, and its item–total correlations were estimated using the Pearson's product moment correlation coefficients between the subscale and total scores of the IMFI-20. A Cronbach's alpha value of ≥0.70 indicates adequate internal consistency [39]. The intraclass correlation coefficient (ICC) was used to assess test–retest reliability within a 1-week interval. An ICC value between 0.75 and 0.90 indicates satisfactory correlation and agreement between two time measurements, and a value of >0.90 indicates excellent correlation and agreement [40].

**2.5.3 Validity.** Exploratory factor analysis (EFA) and principal component analysis [41] with orthogonal rotation [42] were performed to examine construct validity. The Kaiser–Meyer–Olkin (KMO) test and Bartlett's test of sphericity were used to assess the correlation

matrix and sampling adequacy of the IMFI-20 [43, 44]. The KMO index value ranges between 0 and 1, and a value of >0.70 is regarded as acceptable; for Bartlett's test of sphericity, a value of <0.05 is regarded as significant [45].

Furthermore, we assessed the known-group, criterion, and convergent validity of the IMFI-20. Known-group validity was assessed by comparing IMFI-20 scores in individuals with the Indonesian version of the PSQI score of >5 (unfavorable sleep quality) with those with the Indonesian version of the PSQI score of ≤ 5 and by comparing the scores of individuals with a HbA1C level of ≥6.5% with those of individuals with a HbA1C level of <6.5% through independent t tests. To examine criterion validity, the association between the IMFI-20 and the FACIT–Fatigue scale was determined by calculating Pearson's correlation. To assess convergent validity, the relationship of the IMFI-20 with the Indonesian versions of the BDI-II and PSQI was examined by estimating Pearson's correlation.

## 3. Results

### 3.1 General characteristics and fatigue assessment in a health-care setting

In total, 200 patients with T2DM (mean age, 53.26 years) were enrolled into this study. Most of them were female (64.5%) and married (64.5%). Approximately 90% of the participants applied diabetes management treatment strategies involving diet management combined with medication use. Their mean HbA1C level was 7.2%. Among the 200 participants who had T2DM, 51.5% (n = 103) reported feelings of fatigue based on a self-reported questionnaire. The characteristics of the study participants are listed in Table 1. The distribution of the IMFI-20 items was appropriate. As presented in S2 Table, a low percentage (0.5%–24.0%) was found for floor effects (the proportion of participants who obtained the minimum score) and ceiling effects (the proportion of participants who obtained the maximum score).

Regarding fatigue assessment in a health-care setting (Table 2), 33% of the participants who experienced fatigue rarely or never discussed their fatigue with their physicians, and 62% stated that their fatigue had rarely or never been treated by their physicians.

Table 1. Demographic characteristic (N = 200).

| Variables | Total | | Fatigued (n = 103) | | Non-fatigued (n = 97) | | P |
|---|---|---|---|---|---|---|---|
| | n | (%) | n | (%) | n | (%) | |
| Age, mean (SD) [a] | 53.26 | (7.1) | 52.82 | (6.5) | 53.73 | (7.7) | 0.36 |
| BMI, mean (SD) [a] | 24.31 | (3.0) | 24.05 | (3.1) | 24.6 | (2.9) | 0.20 |
| Female [b] | 129 | (64.5) | 62 | (60.2) | 67 | (69.1) | 0.19 |
| Junior high school and above [b] | 115 | (57.5) | 56 | (54.4) | 59 | (60.8) | 0.36 |
| Married [b] | 129 | (64.5) | 66 | (64.1) | 63 | (64.9) | 0.89 |
| Income ≥ 207.08 USD per month [b] | 98 | (49) | 55 | (53.4) | 43 | (44.3) | 0.20 |
| Diabetes duration, mean (SD) [a] | 5.82 | (3.3) | 5.86 | (3.5) | 5.77 | (3.2) | 0.85 |
| Current treatment for diabetes [b] | | | | | | | 0.23 |
| Diet alone | 22 | (11) | 14 | (13.6) | 8 | (8.2) | |
| Diet combined with medication | 178 | (89) | 89 | (86.4) | 89 | (91.8) | |
| Fasting blood glucose (mg/dl), mean (SD)[a] | 122.4 | 14.5 | 129.8 | (12.7) | 114.6 | (12.1) | 0.01 |
| Random blood glucose (mg/dl), mean (SD)[a] | 181.9 | 43.9 | 206.0 | (45.6) | 156.4 | (22.4) | 0.01 |
| HbA1C (%), mean (SD)[a] | 7.2 | 1.5 | 8.1 | (1.5) | 6.1 | (0.7) | 0.01 |

Abbreviations: BMI, Body Mass Index; USD, U.S. Dollar; DM, Diabetes Mellitus; SD, Standard Deviation;

[a], independent t-test;

[b], Chi squared test

**Table 2. Healthcare utilization of fatigue among patient with type 2 diabetes (n = 200).**

| Questions | Total | | Fatigued (n = 103) | | Non-fatigued (n = 97) | |
|---|---|---|---|---|---|---|
| | n | % | n | % | n | % |
| How long have you felt fatigue? (month), mean (SD) | 2.91 | (3.5) | 5.65 | (2.8) | 0.00 | (0.0) |
| Have you ever discussed "the feeling of tired" or "fatigue" with your physician? | | | | | | |
| Never | 61 | (30.5) | 13 | (12.6) | 48 | (49.5) |
| Rarely | 53 | (26.5) | 21 | (20.4) | 32 | (33.0) |
| Sometimes | 69 | (34.5) | 52 | (50.5) | 17 | (17.5) |
| Often | 17 | (8.5) | 17 | (16.5) | 0 | (0) |
| Always | 0 | (0) | 0 | (0) | 0 | (0) |
| Whether your fatigue level has been measured by your physician? | | | | | | |
| Never | 61 | (30.5) | 13 | (12.6) | 48 | (49.5) |
| Rarely | 78 | (39) | 41 | (39.8) | 37 | (38.1) |
| Sometimes | 52 | (26) | 40 | (38.8) | 12 | (12.4) |
| Often | 9 | (4.5) | 9 | (8.8) | 0 | (0) |
| Always | 0 | (0) | 0 | (0) | 0 | (0) |
| Has your fatigue been treated by physician? | | | | | | |
| Never | 67 | (33.5) | 18 | (17.5) | 49 | (50.5) |
| Rarely | 83 | (41.5) | 46 | (44.7) | 37 | (38.1) |
| Sometimes | 44 | (22) | 33 | (32.0) | 11 | (11.4) |
| Often | 6 | (3) | 6 | (5.8) | 0 | (0) |
| Always | 0 | (0) | 0 | (0) | 0 | (0) |

## 3.2 Validity

**3.2.1 Construct validity.** The results of the KMO test and Bartlett's test of sphericity indicated that the number of items in the subscales was adequate, and that the correlation matrix was not an identity matrix (KMO = 0.88; $\chi^2$ = 2414.14, $P < 0.001$). As presented in Table 3, the factor loading of each item was significant ($P < 0.05$) at an acceptable level ($>0.4$). The model with a four-factor structure (i.e., general and physical fatigue, mental fatigue, reduced activity, and reduced motivation) exhibited the most favorable model fit, and this structure was thus extracted; the structure explained 65.81% of the total variance.

**3.2.2 Known-group validity.** Table 4 summarizes the results of the known-group validity analysis. A PSQI (Indonesian version) score of $>5$ corresponded to significantly higher total and subdomain scores for the IMFI-20 relative to a PSQI score of $\leq5$ (all $P < 0.05$). Furthermore, a HbA1C level of $\geq6.5\%$ corresponded to significantly higher total and subdomain scores for the IMFI-20 relative to a HbA1C level of $<6.5\%$ (all $P < 0.05$).

**3.2.3 Criterion validity and convergent validity.** As provided in S3 Table, the total and subscale scores for the IMFI-20 were significantly and negatively correlated with FACIT–Fatigue scale scores ($r = -0.33$ to $-0.71$, all $P < 0.01$), indicating that the IMFI-20 exhibited satisfactory criterion validity for detecting fatigue in patients with T2DM. Regarding convergent validity, the total and subscale scores for the IMFI-20 were significantly and positively associated with the total scores for the Indonesian versions of the BDI-II and PSQI (all $P < 0.01$). Thus, patients with T2DM who experience a higher level of fatigue also experience more severe depression symptoms and poorer sleep quality.

**3.2.4 Reliability.** The Cronbach's alpha value was 0.92 for the total IMFI-20 score and 0.92, 0.82, 0.85, and 0.75 for the scores of the general/physical fatigue, mental fatigue, reduced activity, and reduced motivation subscales. Additionally, the Cronbach's alpha values for each

**Table 3. Exploratory factor analysis of the IMFI-20.**

| Items | Indonesian/English | M ± SD | % Floor effect | % Ceiling effect | Factor Loading | Variance (%) | Eigenvalue |
|---|---|---|---|---|---|---|---|
| General Fatigue/Physical Fatigue | | | | | | 42.40 | 8.48 |
| 1 | Saya merasa bugar / I feel fit | 2.5 ± 1.3 | 23.5 | 12.5 | 0.748 | | |
| 2 | Secara fisik, saya merasa hanya bisa melakukan sedikit / Physically, I feel only able to do a little | 2.6 ± 1.1 | 16.0 | 6.0 | 0.746 | | |
| 5 | Saya merasa Lelah / I feel tired | 2.6 ± 1.2 | 17.0 | 11.0 | 0.741 | | |
| 8 | Secara fisik, saya bisa menangani banyak hal / Physically I can take on a lot | 2.6 ± 1.1 | 18.5 | 8.0 | 0.739 | | |
| 12 | Saya sudah beristirahat / I am rested | 2.5 ± 1.2 | 21.0 | 10.0 | 0.738 | | |
| 14 | Secara fisik, saya merasa dalam kondisi buruk / Physically I feel I am in a bad condition | 2.5 ± 1.1 | 16.5 | 5.5 | 0.698 | | |
| 16 | Saya mudah lelah / I tire easily | 2.6 ± 1.2 | 19.0 | 10.0 | 0.754 | | |
| 20 | Secara fisik, saya merasa dalam kondisi sangat bagus / Physically I feel I am in an excellent condition | 2.7 ± 1.1 | 18.0 | 6.0 | 0.774 | | |
| Mental Fatigue | | | | | | 9.43 | 1.89 |
| 7 | Ketika saya sedang melakukan sesuatu, saya tetap bisa fokus melakukannya / When I am doing something, I can keep my thoughts on it | 2.5 ± 1.1 | 18.0 | 4.5 | 0.768 | | |
| 11 | Saya bisa berkonsentrasi dengan baik / I can concentrate well | 2.6 ± 1.2 | 16.0 | 7.5 | 0.766 | | |
| 13 | Perlu banyak usaha untuk berkonsentrasi pada banyak hal / It takes a lot of effort to concentrate on things | 2.5 ± 1.1 | 17.0 | 7.5 | 0.762 | | |
| 19 | Pikiran saya mudah mengembara / My thoughts easily wander | 2.6 ± 1.2 | 18.0 | 8.5 | 0.760 | | |
| Reduced Activity | | | | | | 8.61 | 1.72 |
| 3 | Saya merasa sangat aktif / I feel very active | 2.0 ± 1.0 | 40.5 | 0.5 | 0.748 | | |
| 6 | Saya pikir, saya melakukan banyak hal dalam sehari / I think I do a lot in a day | 2.1 ± 1.1 | 39.0 | 3.0 | 0.718 | | |
| 10 | Saya pikir, saya melakukan sangat sedikit hal dalam sehari / I think I do very little in a day | 2.1 ± 1.0 | 37.0 | 0.5 | 0.714 | | |
| 17 | Saya menyelesaikan sedikit hal / I get little done | 1.9 ± 0.9 | 44.0 | 6.0 | 0.679 | | |
| Reduced Motivation | | | | | | 5.37 | 1.07 |
| 4 | Saya merasakan ingin melakukan segala macam hal yang menyenangkan / I feel like doing all sorts of nice things | 1.8 ± 0.8 | 36.5 | 0.5 | 0.745 | | |
| 9 | Saya merasa takut untuk melakukan banyak hal / I dread having to do things | 1.7 ± 0.7 | 46.5 | 0.5 | 0.739 | | |
| 15 | Saya punya banyak rencana / I have a lot of plans | 1.8 ± 1.0 | 42.5 | 3.5 | 0.734 | | |
| 18 | Saya tidak ingin melakukan apapun / I don't feel like doing anything | 2.0 ± 0.9 | 25.5 | 3.0 | 0.706 | | |
| Total variance (%) | | | | | | 65.81 | |
| Kaiser-Meyer-Olkin (Bartlett's Test of Sphericity) | 0.875 (p<0.001) | | | | | | |

**Table 4. Known group validity of IMFI-20 for different level of variables (n = 200).**

| Groups | n | IMFI-20 | General/Physical Fatigue | Mental fatigue | Reduced activity | Reduced motivation |
|---|---|---|---|---|---|---|
| PSQI | | | | | | |
| Good sleep quality (≤5) | 167 | 43.0 ± 12.4* | 19.0 ± 6.8* | 9.87 ± 3.7* | 7.35 ± 2.9* | 6.9 ± 2.3* |
| Poor sleep quality (>5) | 33 | 61.6 ± 7.9* | 28.7 ± 6.1* | 11.6 ± 2.9* | 11.6 ± 2.8* | 9.8 ± 3.7* |
| HbA1C | | | | | | |
| < 6.5% | 86 | 36.4 ± 8.7* | 15.2 ± 4.4* | 8.6 ± 2.8* | 5.9 ± 2.2* | 6.7 ± 2.2* |
| ≥ 6.5% | 114 | 53.5 ± 12.1* | 24.6 ± 6.9* | 11.3 ± 3.7* | 9.7 ± 3.1* | 7.8 ± 2.6* |

Values are presented as mean and standard deviation (SD).

*, $p < 0.05$.

item ranged between 0.91 and 0.92. The total IMFI-20 score was significantly correlated with the IMFI-20 subscale scores; the correlation coefficient between the overall scale and each subscale (*r*) was between 0.598 and 0.917 (*P* = 0.001). An ICC value of 0.93 was obtained (95% confidence interval = 0.85–0.96), indicating high similarity between the item scores.

## 4. Discussion

Our study revealed that the prevalence of fatigue in participants was 51.5%, which is similar to the result of a meta-analysis [2]. In addition, the mean duration of fatigue experienced by the participants was 5.65 months. An unanticipated finding is that approximately 33% of the participants who experienced fatigue rarely or never mentioned their fatigue to their physicians; furthermore, 62% reported that their fatigue had rarely or never been treated by their physicians. These findings suggest that, in a clinical setting, health-care providers frequently overlook the fatigue experienced by individuals with T2DM, and that effective interventions for relieving fatigue in individuals with T2DM are lacking. Fatigue in patients with T2DM is associated with glycemic variability and ineffective diabetes self-care [6, 7]. Therefore, health-care providers must design effective strategies for managing the fatigue of patients with T2DM.

A key finding of the present study is its EFA result regarding the four-factor structure of the IMFI-20 in patients with T2DM. This finding corresponds to those of methodological studies examining patients with Parkinson's disease, thyroid disease, and Hodgkin lymphoma [14, 19, 21]; that is, the IMFI-20 cannot distinguish physical fatigue from general fatigue; thus, these two concepts are combined as a subscale for measuring the physical aspect of fatigue in individuals. By contrast, in the study that proposed and developed the original MFI [12] and in another study of Hindi-speaking patients with cancer [17], the five-factor structure of the MFI was adopted. The three-factor structure of the MFI was applied to Chinese-speaking and Polish-speaking patients with cancer [16, 46]. The discrepancy between our findings and previous findings is attributable to the translation of the items pertaining to general fatigue (i.e., items 1, 5, 12, and 16) and physical fatigue (i.e., items 2, 8, 14, and 20), which were assigned the same meaning in terms of physical fitness. Therefore, when the participants rated their physical fitness–related fatigue, those with a lower level of physical fitness may have experienced a higher level of fatigue. Thus, the scores for the items in the general fatigue subscale exhibited a substantial tendency in the assessment of physical fatigue among Indonesian-speaking individuals with T2DM.

The known-group validity results revealed that the participants with poorer sleep quality (PSQI score > 5) tended to experience more severe fatigue. This finding is supported by the result of another study, which reported favorable known-group validity for the MFI-20 in U.S. adults with chronic fatigue syndrome [47]. Poor sleep quality has been recognized as a predictor of fatigue in individuals with T2DM [4], and it has a negative effect on their ability to manage their diabetes and control their blood glucose levels [48, 49]. Partially consistent with another study finding that the fatigue level is associated with the HbA1C level [50], we discovered that the participants who had a HbA1C level of ≥6.5% experienced higher levels of fatigue relative to those with a HbA1C level of <6.5%. However, in contrast to the findings of other studies, the present study did not reveal a significant relationship between the fatigue and HbA1C levels of the participants [8, 51]. The discrepancies between our findings and previous findings indicate that future research on the longitudinal association between fatigue and HbA1C is required.

### 4.1 Limitations and strengths

This study has several limitations. First, it enrolled participants who lived in urban areas and were mostly women, which may limit the generalizability of the results. Second, confirmatory

factor analysis was not performed because of the limited sample size. Future studies should conduct analyses to verify the factorial structure of the IMFI-20. Last, we used self-designed item to determine patients with fatigue or not which may threaten the internal validity of our findings. Nevertheless, the present study achieved a sufficient sample size for investigating the psychometric properties of the IMFI-20 and for using this fatigue assessment framework to assess fatigue in Indonesian-speaking individuals with T2DM in a health-care setting.

## 5. Conclusion

The IMFI-20 is a reliable and valid instrument that comprises four subscales (general and physical fatigue, mental fatigue, reduced activity, and reduced motivation) for assessing specific aspects of fatigue in Indonesian-speaking patients with T2DM. Furthermore, we verified that half of the participants who had T2DM experienced fatigue, and that fatigue-related symptoms were severely neglected by health-care providers. Given these findings, health-care professionals must increase their awareness of fatigue in patients with T2DM and pay attention to the clinical relevance of fatigue-related symptoms in patients with T2DM.

## Supporting information

**S1 Table. STROBE statement—Checklist of items that should be included in reports of** *cross-sectional studies.*
(DOCX)

**S2 Table. Floor and ceiling effects and internal consistency of IMFI-20.**
(DOCX)

**S3 Table. Correlations of IMFI-20 with FACIT-F, BDI-II, and PSQI.**
(DOCX)

**S1 Data.**
(CSV)

## Acknowledgments

We would like to thank all of the participants in this study for their valuable contributions, which we greatly appreciate.

## Author Contributions

**Conceptualization:** Debby Syahru Romadlon, Hsiao-Yean Chiu.

**Data curation:** Debby Syahru Romadlon, Hui-Chuan Huang, Yang-Ching Chen, Sophia H. Hu, Ollyvia Freeska Dwi Marta, Safiruddin Al Baqi, Hsiao-Yean Chiu.

**Formal analysis:** Debby Syahru Romadlon, Faizul Hasan, Hsiao-Yean Chiu.

**Software:** Debby Syahru Romadlon, Faizul Hasan, Hsiao-Yean Chiu.

**Validation:** Debby Syahru Romadlon, Faizul Hasan, Milton D. Chiang Morales, Hsiao-Yean Chiu.

**Visualization:** Debby Syahru Romadlon, Hsiao-Yean Chiu.

**Writing – original draft:** Debby Syahru Romadlon.

**Writing – review & editing:** Debby Syahru Romadlon, Hsiao-Yean Chiu.

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
