## [Decision Letter · Decision Letter 0]

26 Sep 2022

PONE-D-22-23618

Fatigue Following Type 2 Diabetes: Psychometric Testing of the Indonesian Version of the Multidimensional Fatigue Inventory-20 and Unmet Needs in Fatigue

PLOS ONE

Dear Dr. Chiu,

Thank you for submitting your manuscript to PLOS ONE. After careful consideration, we feel that it has merit but does not fully meet PLOS ONE’s publication criteria as it currently stands. Therefore, we invite you to submit a revised version of the manuscript that addresses the points raised during the review process.

Based on the reviewers' suggestions, the paper needs major revision.  The reviewers' comments can be found below.

We look forward to receiving your revised manuscript.

Kind regards,

Tanja Grubić Kezele, Ph.D., M.D.

Academic Editor

PLOS ONE

Journal Requirements:

2. In the ethics statement in the Methods and online submission information, please ensure that you have specified (1) whether consent was informed and (2) what type you obtained (for instance, written or verbal, and if verbal, how it was documented and witnessed). If your study included minors, state whether you obtained consent from parents or guardians. If the need for consent was waived by the ethics committee, please include this information.

Reviewers' comments:

Reviewer's Responses to Questions

Comments to the Author

1. Is the manuscript technically sound, and do the data support the conclusions?

Reviewer #1: Yes

Reviewer #2: Yes

Reviewer #3: Partly

2. Has the statistical analysis been performed appropriately and rigorously?

Reviewer #1: Yes

Reviewer #2: Yes

Reviewer #3: I Don't Know

3. Have the authors made all data underlying the findings in their manuscript fully available?

Reviewer #1: No

Reviewer #2: Yes

Reviewer #3: No

4. Is the manuscript presented in an intelligible fashion and written in standard English?

Reviewer #1: Yes

Reviewer #2: Yes

Reviewer #3: Yes

5. Review Comments to the Author

Reviewer #1: This study examined the psychometric properties of IMFI-20 in Indonesian-speaking patients with T2DM. It showed IMFI-20 is a reliable and valid instrument assessing specific aspects of fatigue in this type of patients. The analysis seems valid; however there were multiple places with ambiguous or even incorrect statements—

1) The last sentence of abstract “We suggest that half of patients with T2DM experience fatigue, a condition is often overlooked by health-care providers, and demonstrated its excellent psychometric properties that can be adopted in studies that use fatigue as an endpoint in the Indonesian-speaking population”— what does “its” refer to? it seems to be IMFI-20; however, in this sentence literally it is not that.

2) The second paragraph of Introduction “… for fatigue in individuals with diabetes, such as …”— I am afraid you mean cancer instead of diabetes.

3) Section 2.5.1 “We applied the independent t test for categorical variables and the chi-squared test for continuous variables to compare demographic characteristics and the fatigue score”— I am afraid it’s switched.

4) Section 3.2.2 “… testing our hypothesis that patients with type 2 diabetes and a score of more than 5 in the Indonesian version of the PSQI or a HbA1C level of 6.5% or more would have a low score in the IMFI-20”— is it really your hypothesis? According to Table 4 it’s the opposite.

Reviewer #2: Comments and Suggestions for Authors

I had the pleasure of reviewing the manuscript " Fatigue Following Type 2 Diabetes: Psychometric Testing of the Indonesian Version of the Multidimensional Fatigue Inventory-20 and Unmet Needs in Fatigue" for PLOS ONE.

Author revealed that half of patients with type 2 diabetes experience fatigue, and that the symptom of fatigue is severely neglected by health-care providers. The study, in my opinion, is interesting, and certainly falls within the scope of the journal. Overall the paper is well written and easy to follow. To my knowledge the topic of this manuscript presents new valuable information on the subject. It is clear that the fatigue is a very important issue that deserves to be studied, especially in in patients with type 2 diabetes, but there are some points in the paper which could deserve further explanations. Most of the manuscript is correct, but there is a need for improvement for the aspects below:

1. The abbreviations throughout the paper requires some attention, for example of T2DM in the Abstract section.

2. Methods are adequate, but it is not clear how the level or cut-off of fatigue was found. Please provide more information of the FACIT-Fatigue scale.

3. I suggest to include the possibility for the floor and ceiling effects in all the MFI-20 subscales as well as for total fatigue score and to expand summary of results and their interpretation.

4. I would suggest including Cronbach's Alpha of all used scales in your study population.

5. Table 4 summarizes the results of IMFI-20 for different level of the PSQI and HbA1C level. State why you used only total IMFI-20 score. I would suggest to include all the MFI-20 subscales as well.

Reviewer #3: The purpose of this paper is to test reliability and validity of the Indonesian version of the MFI-20 in people with type 2 diabetes, and explore its use for fatigue assessment. In general the paper is well-written and meets this objective.

However, one significant omission and concern is the lack of information/data provided on the MFI-20 results itself. The multidimensional fatigue inventory MFI-20 is designed to evaluate 5 dimensions of fatigue, which gives a robust picture of the severity and type of fatigue, but these results are curiously not presented at all in this paper. The individual questions are included in table 3 with factor analysis, it would be easy enough to also include the descriptive statistics for the responses to each question. The authors report that they used the MFI-20 scores to test reliability and validity, including correlations with total and subscale scores between other measures of fatigue which seems appropriate. However, for other analyses it appears that the researchers operationally defined fatigue/no fatigue based on 5 questions that they designed, including a yes/no item "do you feel fatigue". The analysis comparing demographics, measures of diabetes severity, and healthcare utilization uses the yes/no fatigue question rather than the more robust MFI-20 results.

The only other major concern is that there was a hypothesis embedded within the methods that should be presented in the introduction, page 10 "we hypothesized that patients with type 2 diabetes who had a score of more than 5 ... would have a low IMFI-20 score". The results of this hypothesis are not clearly presented later in the paper.

6. PLOS authors have the option to publish the peer review history of their article (what does this mean?). If published, this will include your full peer review and any attached files.

Do you want your identity to be public for this peer review? For information about this choice, including consent withdrawal, please see our Privacy Policy.

Reviewer #1: No

Reviewer #2: Yes: Nijolė Kažukauskienė

Reviewer #3: Yes: Patricia M Kluding

---

## [Author Response · Author response to Decision Letter 0]

27 Oct 2022

Reviewer #1 comments to author:

Reviewer #1: This study examined the psychometric properties of IMFI-20 in Indonesian-speaking patients with T2DM. It showed IMFI-20 is a reliable and valid instrument assessing specific aspects of fatigue in this type of patients. The analysis seems valid; however there were multiple places with ambiguous or even incorrect statements.

Response: Thank you for encouraging us for this project. We have carefully reviewed and revised the manuscript in accordance with your comments and suggestions.

1) The last sentence of abstract “We suggest that half of patients with T2DM experience fatigue, a condition is often overlooked by health-care providers, and demonstrated its excellent psychometric properties that can be adopted in studies that use fatigue as an endpoint in the Indonesian-speaking population”— what does “its” refer to? it seems to be IMFI-20; however, in this sentence literally it is not that.

Response: We have revised the final sentence of the abstract to “Our findings suggest that patients with T2DM who experience fatigue are often overlooked by health-care providers, and that the IMFI-20, which exhibits excellent psychometric properties, can be adopted by studies that use fatigue as an endpoint in Indonesian-speaking populations. All changes are marked in red.

2) The second paragraph of Introduction “… for fatigue in individuals with diabetes, such as …”— I am afraid you mean cancer instead of diabetes.

Response: We have updated the cited references to emphasize that we are referring to patients with diabetes. The paragraph has been revised to “To date, several self-administered measures can be used to evaluate fatigue in individuals with diabetes; they include the Functional Assessment Chronic Illness Therapy (FACIT)–Fatigue scale [10], Fatigue Severity Scale, Fatigue Assessment Scale, and Visual Analog Fatigue Scale [11]. All changes are marked in red.

3) Section 2.5.1 “We applied the independent t test for categorical variables and the chi-squared test for continuous variables to compare demographic characteristics and the fatigue score”— I am afraid it’s switched.

Response: These were typo errors, and we have revised the sentence to “For a comparison of demographic characteristics and fatigue scores, the independent t test was used for evaluating continuous variables and the chi-squared test for categorical variables”. All changes are marked in red.

4) Section 3.2.2 “… testing our hypothesis that patients with type 2 diabetes and a score of more than 5 in the Indonesian version of the PSQI or a HbA1C level of 6.5% or more would have a low score in the IMFI-20”— is it really your hypothesis? According to Table 4 it’s the opposite.

Response: This was a typo error. We have revised the paragraph describing known-group validity to the following: “Table 4 summarizes the results of the known-group validity analysis. A PSQI (Indonesian version) score of >5 corresponded to significantly higher total and subdomain scores for the IMFI-20 relative to a PSQI score of ≤5 (all P < 0.05). Furthermore, a HbA1C level of ≥6.5% corresponded to significantly higher total and subdomain scores for the IMFI-20 relative to a HbA1C level of <6.5% (all P < 0.05)”. All changes are marked in red.

 

Reviewer #2 comments to author:

Reviewer #2: Comments and Suggestions for Authors

I had the pleasure of reviewing the manuscript " Fatigue Following Type 2 Diabetes: Psychometric Testing of the Indonesian Version of the Multidimensional Fatigue Inventory-20 and Unmet Needs in Fatigue" for PLOS ONE.

Author revealed that half of patients with type 2 diabetes experience fatigue, and that the symptom of fatigue is severely neglected by health-care providers. The study, in my opinion, is interesting, and certainly falls within the scope of the journal. Overall the paper is well written and easy to follow. To my knowledge the topic of this manuscript presents new valuable information on the subject. It is clear that the fatigue is a very important issue that deserves to be studied, especially in in patients with type 2 diabetes, but there are some points in the paper which could deserve further explanations.

Response: Thank you for the compliment on our project. Considering your comments and suggestions into account, we revised the manuscript carefully.

Most of the manuscript is correct, but there is a need for improvement for the aspects below:

1. The abbreviations throughout the paper requires some attention, for example of T2DM in the Abstract section.

Response: Thank you for the comment. We have carefully checked and revised the abbreviations of throughout the paper and abstract, particularly T2DM. All changes are marked in red.

2. Methods are adequate, but it is not clear how the level or cut-off of fatigue was found. Please provide more information of the FACIT-Fatigue scale.

Response: We have added information on the scoring system of the FACIT-Fatigue scale in the Methods section; the content added is as follows: “Each item is rated on a response scale, ranging from 0 (not at all) to 4 (very much so). A total score between 0 and 52 is obtained by summing all items, and a higher score indicates less fatigue [11]”. All changes are marked in red.

3. I suggest to include the possibility for the floor and ceiling effects in all the MFI-20 subscales as well as for total fatigue score and to expand summary of results and their interpretation. 

Response: Thank you for the suggestions. We have reported and interpreted the floor effect and ceiling effect of total and subscales scores of the IMFI-20 in the main text (Methods and Results section) as well as S3 Table. All changes were marked in red.

4. I would suggest including Cronbach's Alpha of all used scales in your study population.

Response: Thank you for the suggestion. We have reported Cronbach's Alpha of all used scales (i.e., FACIT–Fatigue scale, Beck Depression Inventory–Second Edition, Pittsburgh Sleep Quality Index) in our study population in the Methods section. Cronbach Alpha for total and subdomain scores of the IMFI-20 were reported in Result section. All changes are marked in red.

5. Table 4 summarizes the results of IMFI-20 for different level of the PSQI and HbA1C level. State why you used only total IMFI-20 score. I would suggest to include all the MFI-20 subscales as well. 

Response: Thank you for the comment and suggestions. We have included the subscales of the in terms of the different level of the PSQI and HbA1C level in Table 4. 

 

Reviewer #3 comments to author:

Reviewer #3: The purpose of this paper is to test reliability and validity of the Indonesian version of the MFI-20 in people with type 2 diabetes, and explore its use for fatigue assessment. In general the paper is well-written and meets this objective.

Response: Your compliments about our project are much appreciated. The manuscript has been revised in response to your comments and suggestions.

However, one significant omission and concern is the lack of information/data provided on the MFI-20 results itself. The multidimensional fatigue inventory MFI-20 is designed to evaluate 5 dimensions of fatigue, which gives a robust picture of the severity and type of fatigue, but these results are curiously not presented at all in this paper. The individual questions are included in table 3 with factor analysis, it would be easy enough to also include the descriptive statistics for the responses to each question. 

Response: Thank you for the comment and suggestion. We have provided descriptive statistics for each question of IMFI-20 in Table 3, including mean and standard deviation, range (minimum-maximum), floor effect, ceiling effect, and internal consistency of total score and all domains of IMFI-20. The descriptive analyses of total and subscale scores are presented in S2 Table. Furthermore, we reverse the sequence of the S2 Table and S3 Table.

The authors report that they used the MFI-20 scores to test reliability and validity, including correlations with total and subscale scores between other measures of fatigue which seems appropriate. However, for other analyses it appears that the researchers operationally defined fatigue/no fatigue based on 5 questions that they designed, including a yes/no item "do you feel fatigue". The analysis comparing demographics, measures of diabetes severity, and healthcare utilization uses the yes/no fatigue question rather than the more robust MFI-20 results. 

Response: We used a self-designed question instead of the MFI-20 results to identify patients with fatigue because the MFI-20 does not provide a clear cut-off point for fatigue. Although we observed significantly different levels of blood glucose and HbA1C between patients with and without fatigue, the unvalidated self-reported item could have decreased the internal validity of our findings. We have highlighted this concern as a study limitation in the following sentence: “Last, we used self-designed item to determine patients with fatigue or not which may threaten the internal validity of our findings.” All changes are marked in red.

The only other major concern is that there was a hypothesis embedded within the methods that should be presented in the introduction, page 10 "we hypothesized that patients with type 2 diabetes who had a score of more than 5 ... would have a low IMFI-20 score". The results of this hypothesis are not clearly presented later in the paper. 

Response: Thank you for the comment. To avoid the confusion, we have removed the hypothesis from the Methods section. In terms of the results regarding the paragraph, we have clearly presented. All changes are marked in red.

---

## [Decision Letter · Decision Letter 1]

11 Nov 2022

Fatigue Following Type 2 Diabetes: Psychometric Testing of the Indonesian Version of the Multidimensional Fatigue Inventory-20 and Unmet Fatigue-Related Needs

PONE-D-22-23618R1

Dear Dr. Chiu,

We’re pleased to inform you that your manuscript has been judged scientifically suitable for publication and will be formally accepted for publication once it meets all outstanding technical requirements.

Kind regards,

Tanja Grubić Kezele, Ph.D., M.D.

Academic Editor

PLOS ONE

Additional Editor Comments (optional):

Reviewers' comments:

Reviewer's Responses to Questions

**Comments to the Author**

1. If the authors have adequately addressed your comments raised in a previous round of review and you feel that this manuscript is now acceptable for publication, you may indicate that here to bypass the “Comments to the Author” section, enter your conflict of interest statement in the “Confidential to Editor” section, and submit your "Accept" recommendation.

Reviewer #1: All comments have been addressed

Reviewer #2: All comments have been addressed

Reviewer #3: (No Response)

2. Is the manuscript technically sound, and do the data support the conclusions?

Reviewer #1: Yes

Reviewer #2: Yes

Reviewer #3: Yes

3. Has the statistical analysis been performed appropriately and rigorously? 

Reviewer #1: Yes

Reviewer #2: Yes

Reviewer #3: Yes

4. Have the authors made all data underlying the findings in their manuscript fully available?

Reviewer #1: No

Reviewer #2: Yes

Reviewer #3: Yes

5. Is the manuscript presented in an intelligible fashion and written in standard English?

Reviewer #1: Yes

Reviewer #2: Yes

Reviewer #3: Yes

6. Review Comments to the Author

Reviewer #1: My comments were addressed, there is no further critique.

My comments were addressed, there is no further critique.

Reviewer #2: The authors report an interesting study about the Psychometric Properties of the Indonesian Version of the Multidimensional Fatigue Inventory-20 in patients with Type 2 Diabetes. To my knowledge the topic of this manuscript presents new valuable information on the subject. The paper has been carefully revised by the authors and they made major revisions improving the quality of the manuscript. I am kindly glad for the sufficiently addressed all the comments. I am in favor of publishing this paper.

Reviewer #3: All questions raised previously have been addressed. Thank you for your efforts on the revision.

7. PLOS authors have the option to publish the peer review history of their article (what does this mean?). If published, this will include your full peer review and any attached files.

Reviewer #1: No

Reviewer #2: **Yes: **Nijolė Kažukauskienė

Reviewer #3: **Yes: **Patricia Kluding PT PhD

---

## [Editor Report · Acceptance letter]

16 Nov 2022

PONE-D-22-23618R1 

Fatigue Following Type 2 Diabetes: Psychometric Testing of the Indonesian Version of the Multidimensional Fatigue Inventory-20 and Unmet Fatigue-Related Needs 

Dear Dr. Chiu:

I'm pleased to inform you that your manuscript has been deemed suitable for publication in PLOS ONE. Congratulations! Your manuscript is now with our production department. 

Kind regards, 

on behalf of

Prof. dr. Tanja Grubić Kezele 

Academic Editor

PLOS ONE